# Game-Based Assessment of Students' Digital Literacy Using Evidence-Centered Game Design [†]

**Jiayuan Li [1], Jie Bai [1], Sha Zhu [1,\*] and Harrison Hao Yang [2,\*]**

[1] National Engineering Research Center for E-Learning, Central China Normal University, Wuhan 430079, China; lijiayuan@mails.ccnu.edu.cn (J.L.); baijie@mails.ccnu.edu.cn (J.B.)

[2] School of Education, State University of New York at Oswego, Oswego, NY 13126, USA

[\*] Correspondence: zhusha@mail.ccnu.edu.cn (S.Z.); harrison.yang@oswego.edu (H.H.Y.)

[†] This paper is an extended version of our paper published in the 16th International Conference on Blended Learning (ICBL), Hong Kong SAR, China, 17–20 July 2023.

**Abstract:** This study measured secondary students' digital literacy using a digital game-based assessment system that was designed and developed based on the Evidence-Centered Game Design (ECGD) approach. A total of 188 secondary students constituted the valid cases in this study. Fine-grained behavioral data generated from students' gameplay processes were collected and recorded with the assessment system. The Delphi method was used to extract feature variables related to digital literacy from the process data, and the Analytic Hierarchy Process (AHP) method was used to construct the measurement model. The assessment results of the ECGD-based assessment had a high correlation with standardized test scores, which have been shown to be reliable and valid in prior large-scale assessment studies.

**Keywords:** digital literacy; digital game-based assessment; ECGD; AHP; assessment model

## 1. Introduction

The rapid development of new generations of information technology, such as artificial intelligence and big data, has had a profound impact on education. As a result, education is currently undergoing a crucial period of digital transformation [1,2]. A key strategy for bridging the digital divide and advancing the digital transformation of education is the enhancement of students' digital literacy [3–6]. Digital literacy comprises the comprehensive ability to effectively use information technology to access and exchange information, create digital content, and adhere to ethical norms in a digital society [7–9]. It is particularly crucial to improve the digital skills and critical thinking of the digital natives born in the 21st century, who are inherently familiar with digital media and resources. The conversion of the digital divide into digital opportunities is of great significance in promoting the advancement of digital education.

Digital literacy assessment is crucial for accurately gauging students' level of digital literacy and is thus a fundamental prerequisite for enhancing their digital literacy skills [10,11]. However, the current methods used to assess students' digital literacy have several limitations in terms of evaluation content and tools, making it a challenge to comprehensively, accurately, and objectively evaluate students' proficiency. First, evaluation content to date has primarily focused on traditional "test question-answer" paradigms [12], emphasizing students' basic cognitive abilities such as digital knowledge and application skills. Consequently, such an approach fails to evaluate higher-order cognitive aspects, such as the ability to use digital technology to analyze and solve problems, engage in innovation and creativity, and address ethical and moral issues in digital society [13,14]. Second, the evaluation tools predominantly rely on standardized tests and self-reported scales [15–18]. While these methods provide a summarized evaluation of students' digital literacy, they are not suited to measuring implicit cognitive abilities and thinking processes [19]. Although several

studies have introduced situational task assessment and portfolio assessment as alternative tools for process evaluation in recent years, the process data collected by the situational task assessment is of single type and coarse granularity, which is relatively inadequate in providing comprehensive evidence reflecting students' digital literacy [20,21]. Establishing scientific and objective evaluation standards for portfolio assessment is challenging; it requires teachers to invest a great deal of time and effort, which limits the application of portfolio assessment in evaluating students' digital literacy [22,23]. Therefore, these two assessment methods could not accurately evaluate students' digital literacy [24]. Given the significance of digital literacy for fostering innovative talent and preparing students to tackle future opportunities and challenges, it is imperative to address the pressing issue of how to accurately evaluate students' digital literacy.

The application of the Evidence-Centered Game Design (ECGD) approach in assessment design has the potential to overcome the limitations of the current methods for the evaluation of students' digital literacy. In the era of artificial intelligence, it is possible to comprehensively and non-invasively record students' online activities. Researchers have thus begun to explore assessments based on process data, guided by the ECGD theory [25]. The ECGD concept emphasizes the use of complex tasks to elicit students' ability and performance. During student evaluation, process data are collected through gamified tasks that are engaging and interactive. This data extraction process captures evidence that reflects students' ability and performance. Building upon this foundation, this study proposes a new method for evaluating students' digital literacy based on the ECGD paradigm. The research team has developed a game-based assessment tool to measure students' digital literacy, which collects fine-grained procedural data generated by students during the task completion process. The Delphi method and the Analytic Hierarchy Process (AHP) are employed to determine the extracted characteristic variables and their weights. Furthermore, the research team has conducted a practical analysis to validate the proposed method [26]. The results of this study will contribute to the iterative optimization of each aspect of the evaluation method, ultimately providing a more reliable and effective approach to accurately assess students' digital literacy levels.

## 2. Literature Review

### 2.1. Digital Literacy

The ideological source of digital literacy can be traced back to the 1995 book Being Digital, by American scholar Nicholas Negroponte. He pointed out that people should become masters of digital technology, able to use digital technology to adapt, participate in, and create digital content for learning, work, and life [27]. With the continuous development of society, the concept of digital literacy has been continuously extended and expanded, in an evolutionary process that can be roughly divided into three stages:

2.1.1. Stage 1: The Period Spanning from the 1990s to the Early 21st Century

The concept of digital literacy was first proposed by Paul Gilster, an American scholar of space science and technology, in his 1997 monograph, "Digital Literacy". He defined digital literacy as "the ability to understand information, and more importantly, the ability to evaluate and integrate various formats of information that computers can provide" [28]. In 2004, the Israeli scholar Yoram Eshet-Alkalai pointed out that digital literacy is regarded as a necessary survival skill in the digital era. It includes the ability to use software and operate digital devices; various complex cognitive, motor, sociological, and emotional skills used in the digital environment; and the ability to perform tasks and solve complex problems in the digital environment [29]. Thus, during Stage 1, people focused on the digitalization of the living environment and the basic skills required to meet the challenges of digitalization. The concept of digital literacy in this period mainly emphasized the ability to understand, use, evaluate, and integrate the digital resources and information provided by computers.

2.1.2. Stage 2: The Period Spanning from the Beginning of the 21st Century to the 2010s

With the real arrival of the digital age, information technology as represented by the Internet accelerated the development of the digital society, and countries around the world began to pay attention to the core ability of digital literacy. In 2003, the digital horizon report issued by the New Zealand Ministry of Education pointed out that digital literacy is a kind of "life skill" that supports the innovative development of ICT in industrial, commercial, and creative processes. Learners need to acquire confidence, skills, and discrimination in order to use information technology in an appropriate way [30]. In 2009, Calvani proposed that digital literacy consists of the ability to flexibly explore and respond to new technological situations; the ability to analyze, select, and critically evaluate data and information; the ability to explore technological potential; and the ability to effectively clarify and solve problems [31]. In Stage 2, the concept of digital literacy emphasized the correct and appropriate use of digital tools—not limited to computers—and attached importance to innovation and creation, emphasizing that users are not only the users of digital technology, but also the producers of digital content.

2.1.3. Stage 3: The Period Spanning from the 2010s to the Present

With the rapid development of artificial intelligence, big data, the Internet of Things, and other information technologies, digital literacy has drawn increasing attention at the national level. At the same time, a range of international organizations have also begun to develop concepts and frameworks of digital literacy, such that its conceptual development can be said to have entered a period of "contention of a hundred schools of thought". For example, in 2018, UNESCO defined digital literacy as "the ability to access, manage, understand, integrate, communicate, evaluate and create information safely and appropriately through digital technologies for employment, decent jobs and entrepreneurship" [32]. In 2021, China proposed that "digital literacy and skills are the collection of a series of qualities and abilities such as digital acquisition, production, use, evaluation, interaction, sharing, innovation, security and ethics that citizens in a digital society should have in their study, work and life" [33]. During this period, the concept of digital literacy focused more on the unity of humans' own development and social advancement, the humanistic attributes of digital literacy, and ethics, morality, laws, and regulations.

In summary, from the conceptual development of digital literacy, the connotation of digital literacy has expanded from only including understanding, using, assessing, and integrating digital resources to innovation and creativity, ethics and morality, etc., and from only proposing cognitive level requirements, such as awareness and knowledge to behavioral and value level requirements [34–36]. In addition, with the continuous development of artificial intelligence, digital literacy ultimately includes a wider range of higher-order content, such as information thinking. The concept of digital literacy has become more comprehensive and essential for individuals to thrive in the digital age. Specifically, digital literacy includes lower-order cognitive aspects such as basic information knowledge, as well as higher-order cognitive aspects such as using information technology to solve problems [37,38]. Based on the above points of view, our research team posits that digital literacy is a multifaceted construct comprising an individual's awareness, ability, thinking and cultivation to properly use information technology to acquire, integrate, manage, and evaluate information; understand, construct, and create new knowledge; and to discover, analyze, and solve problems [39].

### 2.2. Digital Literacy Framework

At present, international organizations and researchers pay great attention to the assessment of students' digital literacy, and have conducted extensive and in-depth research on the establishment of the assessment framework of students' digital literacy. For example, the European Commission (EC) proposed DigComp1.0 [40] in 2013, including five areas of digital competence: information, communication, content-creation, safety, and problem-

solving. In 2016, EC proposed DigComp2.0 on the basis of DigComp1.0 [41], including five competence areas: information and data literacy, communication and collaboration, digital content creation, safety, and problem solving. UNESCO released a digital literacy global framework [32] in 2018, adding two new dimensions: "hardware and software operations" and "career related competencies" on the basis of DigComp2.0. Additionally, in terms of researchers, Eshet-Alkalai proposed a digital literacy framework based on years of research and practical experience, including photo-visual literacy, reproduction literacy, branching literacy, information literacy, and socio-emotional literacy [29].

Based upon the review of the digital literacy framework proposed by international organizations and researchers, the research team extracted, sorted, and merged the key words of the assessment indicators of students' digital literacy. Thus, the assessment framework of students' digital literacy is proposed, which includes four dimensions: information awareness and attitude (IAA), information knowledge and skills (IKS), information thinking and behavior (ITB), and information social responsibility (ISR) [42]. IAA refers to the sensitivity, judgment and protection awareness of information privacy and security, including information perception awareness, information application awareness, and information security awareness; IKS mainly investigates students' understanding of information science knowledge, mastering, and using common software or tools to complete the creation of digital works, mainly involving information science knowledge and information application skills; ITB mainly refers to the thinking ability to abstract, decompose, and design problems in daily life and learning situations, as well as the comprehensive consciousness and ability tendency to carry out digital communication and learning, mainly including two levels of information thinking and information behavior; ISR refers to the responsibility of students' activities in the information society in terms of laws, regulations, and ethics, including information ethics and laws and regulations [39].

*2.3. The Digital Literacy Assessment*

Initially, the evaluation of students' digital literacy was mainly based on classical test theory (CTT) [43,44]. Researchers compiled or adapted digital literacy evaluation tools (mainly self-reported scales) through a literature review and expert consultation, and used them to measure students' digital literacy level. This method mainly determines students' digital literacy level based on their reports. Thus, the results are greatly affected by subjective factors in the participants, and, thus, there is a need for improvement in the evaluation accuracy of this method [45–48]. In view of the limitations of CCT and the strong subjectivity of the evaluation results of self-reported scales, the academic community began to explore the use of item response theory (IRT) in the evaluation of students' digital literacy [49]. At this stage, evaluation tools were mainly based on standardized test questions. The evaluation of students' digital literacy based on item response theory provides a unified standard for measuring both the subjects' digital literacy level and the statistical parameters of the items. This approach successfully addresses issues such as the difficulty of the estimation of various parameters depending too much on the case, and the subjectivity of the evaluation results, thus effectively enhancing the accuracy of the evaluation results. For example, Zhu developed a standardized test measuring students' digital literacy, comprising 37 multiple-choice questions using a Rasch model based on IRT, yielding a more accurate and objective tool for the evaluation of students' digital literacy [50]. Later, Nguyen and Habók also adopted a similar approach and compiled a digital literacy test comprising self-reported scales and standardized test questions, in which multiple-choice questions were used to measure the students' digital knowledge [19]. However, relying solely on such self-reported scales and standardized tests is still problematic for meeting the need to evaluate high-order thinking abilities such as computational thinking, digital learning, and innovation in digital literacy [51,52]. Moreover, such tools yield summative evaluations, and the evaluation results cannot well reflect the actual level of students' digital literacy [53,54].

*2.4. Game-Based Assessment of Digital Literacy Based on ECGD*

Mislevy proposed the ECGD approach [55]. This is based on the Evidence-Centered Design (ECD) approach and incorporates game-based tasks into assessment design. ECD is a systematic method that guides the design of educational evaluations [56]. Compared to traditional assessment methods—such as CTT, which assigns values to potential traits based on features—ECD may collect more extensive and fine-grained process data from students, which allows for a comprehensive and accurate evaluation of their complex implicit abilities. Specifically, ECD emphasizes the construction of complex task situations, obtaining multiple types of procedural data, and achieving evidence-based reasoning [57]. At present, ECD has been widely used in international large-scale assessment programs, such as PISA, NAEP, and ATC21S, which were designed and developed based on the ECD evaluation framework. In addition, ECD is also widely used in the evaluation of core literacy, 21st-century skills, computational thinking, data literacy, logical reasoning ability, scientific literacy, and other forms of high-level thinking. Some researchers have also used ECD to assess students' digital literacy. For example, Avdeeva evaluated the digital skills dimension of students' digital literacy using a method based on ECD [58]. They verified the advantages of ECD theory in terms of accuracy and other aspects, as compared to IRT. Zhu conducted an in-depth analysis of ECD theory, were the first to put forward the idea of students' digital literacy evaluation based on ECD, and initially constructed an evaluation method for students' digital literacy driven by theory and data [42].

The advantage of game-based assessment tasks is that one can create an environment of richness, playability, and simulation. Doing so can effectively reduce anxiety in the assessment process and improve participation, and is suitable for assessing students' higher-order cognitive aspects [59]. In addition, a game-based assessment task can also establish a task set with multiple difficulty levels to assess students' performance in different situations, so as to effectively distinguish the assessment results [60]. A game-based assessment task can be a continuous or cyclic process, which can obtain students' procedural behavior data [61]. The ECGD conceptual framework is the basis of evaluation design, including three main models: the student model, the task model, and the evidence model. The student model defines the knowledge, skills, and abilities (KSAs) to be measured. The evidence model describes how to update the information for student variables in the task model based on the test-takers' performance in the task. The task model describes how to structure different kinds of situations to evoke student performance to obtain data. Evaluation methods based on ECGD emphasize the creation of complex and realistic tasks to arouse students' performance on KSA, which helps to reflect students' digital literacy in real scenes. Students' game processes can also generate rich and complex process data, which can provide rich evidence to reflect students' KSA.

To date, many researchers have used ECGD to evaluate higher-order cognitive aspects. For example, Chu and Chiang developed a game-based assessment tool based on ECGD to measure scientific knowledge and skills [62]. The results show that task-related behavior characteristics are an effective means of predicting students' mastery of overall skills. Bley developed a game-based assessment system that uses the ECGD method to measure the internal capabilities of enterprises [63]. The results showed that learners' entrepreneurial ability and cognitive performance in tasks can be accurately measured in this way. However, there have been few empirical studies on digital literacy assessment using ECGD. Although our research team has previously developed a game-based assessment of students' digital literacy based on ECGD [64], the research only proposed a conceptual framework and designed a game-based assessment tool for students' digital literacy, but did not verify its effectiveness using empirical measurement data. Therefore, this study aimed to evaluate secondary school students' digital literacy based on the conceptual framework of ECGD, so as to verify the effectiveness of our ECGD-based assessment approach.

## 3. Methods

### 3.1. Participants

This study was conducted with five classes of a middle school located in Wuhan Economic Development Zone, China. The selected school and classes were chosen for the following reasons: (1) the school highly prioritizes the cultivation of students' digital literacy, and has actively participated in prior digital literacy assessments for students; (2) the students were familiar with the regulations and processes of computer-based assessment and showed enthusiasm for participating in the game-based assessment. A total of 210 seventh-grade students, comprising 114 boys and 96 girls, participated in this study. The students' ages ranged from 12 to 15 years old. Prior to participation, all participants were provided with information regarding the study's purpose and were required to sign a formal consent form in order to participate.

### 3.2. Instruments

3.2.1. A Digital Game-Based System for Assessing Students' Digital Literacy

In order to induce performance relative to the students' digital literacy ability model, this study uses the narrative game "Guogan's Journey to A Mysterious Planet", developed by our research team to evaluate students' digital literacy [64]. The game consists of a total of 13 tasks measuring the four dimensions of students' digital literacy, which can be categorized into five different types: multiple-choice question, maze, dragging question, matching question, and sorting question. Each task was designed to measure one dimension of students' digital literacy. For instance, Figure 1 shows a dragging question, which requires the students to discover patterns or rules by analyzing the existing information and drag the correct color block to the right position. This task examines the ITB dimension of students' digital literacy. Students are given two chances to complete each task. If a student finishes a task incorrectly twice, the system presents a "Pass card" and begins the next task. During gameplay, students gain gold coins based on their task performance. Students can choose whether to click a "Help" button during the gameplay process; doing so costs them coins, but it will provide students with some hints to complete the tasks successfully. They can also click a "Return" button to return to the previous page to confirm information.

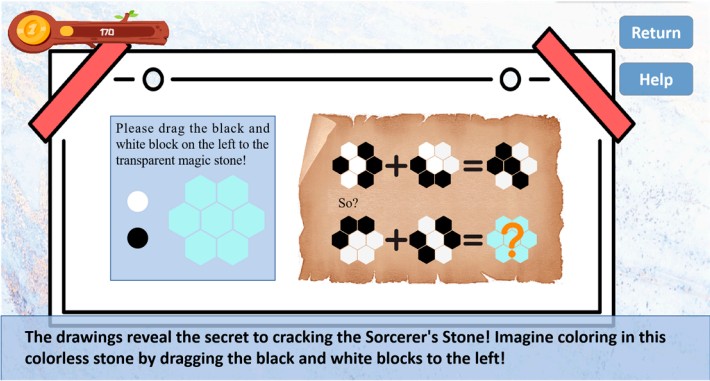

**Figure 1.** Example image of dragging question.

Table 1 shows the details of the game-based assessment tasks, including the task type, the corresponding dimension of digital literacy, and the observed variables. These observable variables have been considered as evidence that can reflect the performance of students' digital literacy through extensive literature reviews and expert consultation [64]. The observed variables include: completion time (the time period elapsed from the player starting the task to completing it), thinking time (the total time the mouse cursor stayed in different areas of the interface during the students' answering process), correctness (whether the task was completed correctly or not), answer counts (number of times of a

task completion), help times (number of times the "Help" button was clicked), return times (number of times the "Return" button was clicked), similarity (the similarity of the action sequence with the reference sequence), and efficiency (the efficiency of the action sequence).

**Table 1.** Description of the game-based assessment tasks.

| Task | Task Type | Dimension | Observed Variables |
|---|---|---|---|
| Tasks 1, 3, 6, 10, 11 | | IAA | completion time, thinking time, correctness, answer counts, help times, return times |
| Tasks 2, 13 | Multiple-choice | ISR | |
| Task 7 | question | IKS | |
| Task 8 | | ITB | |
| Task 4 | Maze | ITB | completion time, thinking time, correctness, answer counts, help times, return times, similarity, efficiency |
| Task 5 | Dragging question | ITB | |
| Task 9 | Matching question | ITB | |
| Task 12 | Sorting question | IKS | |

### 3.2.2. Standardized Test

To verify the results of the game-based assessment test, this study used the digital literacy standardized test designed in our previous study, comprising 24 multiple-choice questions and multiple-answer questions (i.e., multiple choice questions with more than one correct answer). For example, the following is a sample question measuring students' IKS: The vehicle position tracker can determine the location of the vehicle in real time. What technology does it use? A. IoT technology; B. big data technology; C. cloud computing technology; D. multimedia technology. This test has been validated many times in China's large-scale student digital literacy assessment project, and has been demonstrated to have good reliability and validity, difficulty, discrimination, and other indicators, with high standard values [65].

### 3.3. Data Collection

With the help of the research team and school administrators, the students from the five classes participated in the digital literacy assessment in the designated computer laboratory. The students were required to complete the game assessment and standardized test within 40 min. Specifically, the assessment procedure comprised three steps. First, before the assessment, the information technology teacher informed the students of the purpose of the evaluation and emphasized the operating rules, browser settings, and other precautions, and distributed the assessment hyperlink to the students through the teacher's computer. Second, upon accessing the assessment hyperlink, students were required to fill in their personal information and complete the digital literacy standardized test. Finally, upon submitting their responses to the standardized test, students were automatically redirected to the digital literacy game-based assessment system, where they completed the game-based assessment tasks according to the situational sequence.

### 3.4. Data Storage

The game-based assessment system uses xAPI to record the process data generated during the assessment. xAPI is a standard for describing and exchanging learning experiences [66,67]. It records the behavior (verb), the object of the behavioral operation (object), the tool used (tool), and the timestamp of the occurrence of the student's (actor) behavior within a context (context), with the task serving as the core. The xAPI data collection framework is employed to characterize the click behavior of students when they complete tasks using specific format statements [68]. These statements are subsequently placed in the learning recording system (LRS) to facilitate the real-time tracking, collection, and storage of students' click data. To generate statements, xAPI specifies the format object representation (JSON) of JavaScript. Figure 2 shows an example of xAPI-based data stored in JSON format.

```
[
    { "actor_id": 16522645142339,
        "context":
                [{"question_id": "21",
                    "task":
                        [{
                            "verb": "enter",
                            "object": "",
                            "tool": "",
                            "timestamp": "1652264613891"
                        },
                        {
                            "verb": "start_time",
                            "object": "1843",
                            "tool": "",
                            "timestamp": "1652264615734"
                        },

                        {
                            "verb": "help_button_click_times",
                            "object": "0",
                            "tool": "",
                            "timestamp": "1652264615736"
                        },
                        {
                            "verb": "total_coins_record",
                            "object": "150",
                            "tool": "",
                            "timestamp": "1652264615737"
                        }]
                "actor_data":

                        {   "qid": 232,
                            "school": "",
                            "tool": "6",
                            "class": "5"
                            "name": "Meng",
                            "gender": "1"
                        },
            },
```

**Figure 2.** An example of xAPI-based data stored in JSON format.

*3.5. Data Pre-Processing and Analysis*

The game-based assessment data of 210 students were collected and matched with their standardized test scores based on the xAPI-based process data. During the assessment, a few students made mistakes or did not follow the operation rules of the game (for example, students accidentally exited the assessment system interface and re-logged in to continue, resulting in abnormal data), which led to the data collected in this study were redundant, inconsistent, and even noisy, and made it difficult to meet the experimental requirements. Therefore, the data were cleaned and preprocessed as follows. (1) Missing data processing. When the proportion of missing student data cases was relatively high, it was necessary for the data to be eliminated. In this study, the cases with a high rate of missing fields were eliminated, and the cases with low missing rates were filled by the mean method. A total of 10 cases were eliminated for this reason. (2) Abnormal data processing. Abnormal data were those generated by students who failed to adhere to the assessment rules for their operations, as this has an impact on the accuracy of the model prediction. A total of

12 cases were eliminated due to human errors, repeated answers by the same student, or too short of a response time. Ultimately, a total of 22 cases were eliminated, and the remaining 188 cases were used as experimental data.

The data analysis phase includes the following five steps: (1) the feature variables of students' digital literacy was determined based on Delphi method; (2) the weight of the feature variables were calculated using the AHP; (3) the game-based assessment data was analyzed using a Rasch model to validate the task design; (4) the results of the game-based assessment for students' digital literacy was analyzed; (5) the effectiveness of the game-based digital literacy assessment method was verified using Spearman correlation analysis.

Firstly, the Delphi method was used to determine the feature variables and construct the evaluation model of students' digital literacy. An "expert advisory group" was established, composed of 14 scholars, teachers, and researchers in the field of digital literacy assessment. They were invited to complete a survey, determine the feature variables of each task in relation to students' digital literacy, and rank these feature variables according to their importance. Specifically, this study conducted three rounds of expert consultations. In each round, a questionnaire was distributed to the experts and their feedback was collected, analyzed, and summarized. The feature variables were revised and iterated based on the experts' opinions. After the third round of consultation, the experts' opinions were generally consistent and met the expected requirements.

Secondly, the weights of the characteristic variables were determined using the AHP. The analysis steps of AHP were as follows: (1) a judgment matrix was built by comparing the importance of the characteristic variables; (2) the weight of the characteristic variable of each task was calculated; and (3) the consistency of the judgment matrix was verified [69–71]. If the consistency of the judgment matrix was verified, then these weights were served as the scoring criteria for measuring students' digital literacy. Otherwise, the researcher would delete the questionnaire data that are inconsistent with the judgment matrix and recalculate the weights [72].

Thirdly, in order to explore the relationship between item difficulty and students' ability, this study generated an item-person map using a Rasch model. This is a graphical representation of person–abilities and item-difficulties, drawn based on the equal measures (logits) of the raw item difficulties and raw person scores [73]. The item–person map is divided into four quadrants, in which person estimates and item estimates are distributed on the left and right sides, respectively, based on person–ability and item-difficulty estimates [74]. Generally, the persons in the upper left quadrant show better abilities, implying that the easier items were not difficult enough for them. Meanwhile, the items on the upper right show higher difficulty, suggesting that they are beyond the students' ability level [50]. Rasch model analysis includes the following three steps: (1) computing the uni-dimensionality; (2) calculating person and item reliability coefficients; and (3) generating an item-person map [75].

Fourthly, the average scores for students' overall digital literacy and its four individual dimensions, as well as the average values of the seven feature variables of each dimension of students' digital literacy, were calculated within the context of the game-based assessment tasks.

Finally, in order to verify the effectiveness of the game-based digital literacy assessment method proposed in this study, a Spearman correlation analysis was conducted to analyze the correlation between the game-based assessment results and the standardized test results. The standardized test utilized by this research team has been applied in multiple large-scale assessments, confirming its reliability and validity as a tool for evaluating students' digital literacy [65]. Therefore, the correlation between the game-based assessment results and the standardized test results could validate the effectiveness of the assessment method proposed in this study.

## 4. Results

### *4.1. Construction of the Evaluation Model for Digital Literacy*

#### 4.1.1. Determination of the Feature Variables

After three rounds of expert opinions, the feature variables of each task were finally formed, as shown in Table 2. Return times was regarded as an inapplicable characteristic variable for all tasks, while completion time, thinking time, and correctness were identified as characteristic variables of tasks 1, 2, 3, 11, and 13. With regard to tasks 5, 9, and 12, all the feature variables other than return times were considered to be relevant variables for assessing students' digital literacy.

**Table 2.** The feature variables of each task as identified by experts.

| Feature Variables / Task | CT | TT | CO | HT | AC | EF | SI | RT |
|---|---|---|---|---|---|---|---|---|
| Task 1 | √ | √ | √ | × | × | × | × | × |
| Task 2 | √ | √ | √ | × | × | × | × | × |
| Task 3 | √ | √ | √ | × | × | × | × | × |
| Task 4 | √ | √ | × | √ | × | √ | √ | × |
| Task 5 | √ | √ | √ | √ | √ | √ | √ | × |
| Task 6 | √ | √ | √ | √ | √ | × | × | × |
| Task 7 | √ | √ | √ | √ | √ | × | × | × |
| Task 8 | √ | √ | √ | × | √ | × | × | × |
| Task 9 | √ | √ | √ | √ | √ | √ | √ | × |
| Task 10 | √ | √ | √ | √ | √ | × | × | × |
| Task 11 | √ | √ | √ | × | × | × | × | × |
| Task 12 | √ | √ | √ | √ | √ | √ | √ | × |
| Task 13 | √ | √ | √ | × | × | × | × | × |

NOTE: CT: completion time; TT: thinking time; CO: correctness; HT: help times; AC: answer counts; EF: efficiency; SI: similarity; RT: return times.

#### 4.1.2. Calculation of the Weight of Feature Variables

After determining the feature variables, this study employed the Analytic Hierarchy Process to calculate the weights of the feature variables of each task. First, a hierarchical structure model of each task's feature variables was constructed. The "expert advisory group" then assessed the relative importance of each task's feature variables by ranking them on a scale from 1 to 9. A judgment matrix was then constructed to calculate the weight of each task's feature variables. The average random consistency index RI was in the range of 0 to 1.59, and the calculated consistency ratio Cr was less than 0.1 [71,76]. All judgment matrices thus met the consistency requirements. Table 3 shows the calculation results for the weights of the feature variables of each task. The weight calculation results indicate that CO had the largest weight proportion across all tasks, while the feature variables with the smallest weights varied across different tasks. For example, the feature variable with the smallest weight in tasks 1, 2, 3, 4, 6, 7, 8, 10, 11, and 13 was TT, while the feature variables with the smallest weights in tasks 5, 9, and 12 were AT, HT, and CT.

**Table 3.** Calculation results of the weights of the feature variables for each task.

| Feature Variables / Task | CT | TT | CO | HT | AC | EF | SI |
|---|---|---|---|---|---|---|---|
| Task 1 | 0.254 | 0.114 | 0.632 | / | / | / | / |
| Task 2 | 0.263 | 0.119 | 0.618 | / | / | / | / |
| Task 3 | 0.276 | 0.118 | 0.606 | / | / | / | / |
| Task 4 | 0.162 | 0.130 | / | 0.266 | / | 0.291 | 0.151 |

**Table 3.** *Cont.*

| Task | CT | TT | CO | HT | AC | EF | SI |
|---|---|---|---|---|---|---|---|
| Task 5 | 0.138 | 0.112 | 0.220 | 0.089 | 0.106 | 0.168 | 0.167 |
| Task 6 | 0.104 | 0.069 | 0.422 | 0.173 | 0.232 | / | / |
| Task 7 | 0.110 | 0.071 | 0.432 | 0.181 | 0.206 | / | / |
| Task 8 | 0.215 | 0.131 | 0.441 | / | 0.213 | / | / |
| Task 9 | 0.131 | 0.106 | 0.242 | 0.103 | 0.113 | 0.149 | 0.156 |
| Task 10 | 0.139 | 0.074 | 0.427 | 0.190 | 0.170 | / | / |
| Task 11 | 0.241 | 0.118 | 0.641 | / | / | / | / |
| Task 12 | 0.101 | 0.109 | 0.262 | 0.094 | 0.114 | 0.168 | 0.152 |
| Task 13 | 0.237 | 0.126 | 0.637 | / | / | / | / |

### 4.1.3. Construction of the Assessment Model

Based upon the weights of the digital literacy evaluation indicators for primary and secondary students as obtained in our previous study [48], and also upon the weighting results obtained from the above analysis, the linear mathematical expression of the digital game-based digital literacy assessment model was calculated as follows:

$$Y = 0.295 \times Y1 + 0.170 \times Y2 + 0.289 \times Y3 + 0.246 \times Y4 \tag{1}$$

where,

$$
\begin{aligned}
Y1 = {} & 0.20 \times (B1 \times 0.254 + B2 \times 0.114 + B3 \times 0.632) + 0.20 \times (B1 \times 0.276 + B2 \times 0.118 \\
& + B3 \times 0.606) + 0.20 \times (B1 \times 0.104 + B2 \times 0.069 + B3 \times 0.422 + B4 \times 0.173 + B5 \\
& \times 0.232) + 0.20 \times (B1 \times 0.139 + B2 \times 0.074 + B3 \times 0.427 + B4 \times 0.190 + B5 \times 0.170) \\
& + 0.20 \times (B1 \times 0.241 + B2 \times 0.118 + B3 \times 0.641)
\end{aligned}
\tag{2}
$$

$$
\begin{aligned}
Y2 = {} & 0.5 \times (B1 \times 0.110 + B2 \times 0.071 + B3 \times 0.432 + B4 \times 0.181 + B5 \times 0.206) + 0.5 \\
& \times (B1 \times 0.101 + B2 \times 0.109 + B3 \times 0.262 + B4 \times 0.094 + B5 \times 0.114 + B6 \times 0.168 + \\
& B7 \times 0.152)
\end{aligned}
\tag{3}
$$

$$
\begin{aligned}
Y3 = {} & 0.25 \times (B1 \times 0.162 + B2 \times 0.130 + B4 \times 0.266 + B6 \times 0.291 + B7 \times 0.151) + \\
& 0.25 \times (B1 \times 0.138 + B2 \times 0.112 + B3 \times 0.220 + B4 \times 0.089 + B5 \times 0.106 + B6 \times \\
& 0.168 + B7 \times 0.167) + 0.25 \times (B1 \times 0.215 + B2 + 0.131 + B3 \times 0.441 + B5 \times 0.213) \\
& + 0.25 \times (B1 \times 0.131 + B2 \times 0.106 + B3 \times 0.242 + B4 \times 0.103 + B5 \times 0.113 + B6 \\
& \times 0.149 + B7 \times 0.156)
\end{aligned}
\tag{4}
$$

$$
Y4 = 0.5 \times (B1 \times 0.263 + B2 \times 0.119 + B3 \times 0.618) + 0.5 \times (B1 \times 0.237 + B2 \times 0.126 + B3 \times 0.637)
\tag{5}
$$

In the above expressions, $Y1$, $Y2$, $Y3$, and $Y4$ represent the four dimensions of digital literacy, namely IAA, IKS, ITB, and ISR; $B1$–$B7$ represent the values of the seven feature variables, namely completion time, thinking time, correctness, help times, answer counts, efficiency, and similarity.

Among these, the values for thinking time and completion time needed to be processed and then weighted. According to the experts' suggestions, the data for thinking time and completion time were processed as follows. (1) Reasonable time ranges for students' thinking times and answering times were determined based on the corresponding time distributions. (2) The duration of thinking times and completion times were divided into four parts (10%, 40%, 70%, and 100% of the duration of the longest time). (3) Values were assigned to different durations, according to the principle that the longer the duration, the lower the score. For example, if a student's thinking time fell within the top 10% of the reasonable time range, then a value of 1 was assigned; if a student's thinking time fell within 100% of the reasonable time range, then a value of ¼ was assigned. Additionally, a value of 1 or 0 was assigned to the variable of correctness, depending on whether the student answered correctly or incorrectly; a value of 1 was assigned to the variable of help only if the students did not click the "Help" button, otherwise 0 was assigned; a value of 1

was assigned to the variable of answer counts only if the student answered correctly the first time, otherwise 0 was assigned. The values of similarity and efficiency were calculated using the Levenshtein distance.

### 4.2. Analysis of the Game-Based Assessment Tasks

This study analyzed the data for the students' digital literacy game-based assessment tasks using a Rasch model. As shown in Figure 3, on the right side, the difficulty of the items decreases from top to bottom; similarly, students' ability decreases from top to bottom on the left side. Item difficulty covered about five logit, while the students' ability covered seven logit. Among them, the most difficult task was task 9, while the easiest task was task 4. The average value of the students' ability in digital literacy was slightly higher than that of the items' difficulty, indicating that the difficulty of the game-based assessment tasks were appropriate to the students' actual level of digital literacy.

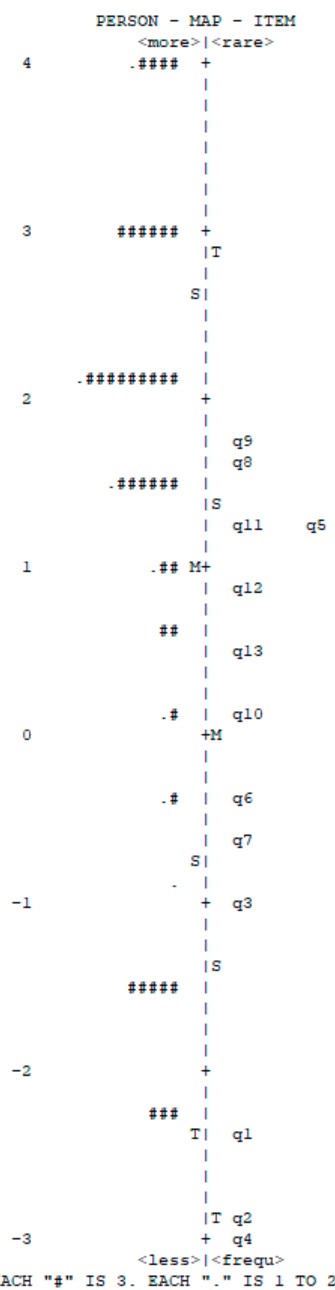

**Figure 3.** The person–item map.

### 4.3. Description of the Digital Game-Based Assessment Results

4.3.1. Analysis of Overall Results for the Game-Based Assessment

Based on the above evaluation model, the average scores for students' overall digital literacy and its four individual dimensions were calculated. The results (see Figure 4) show that the students' overall digital literacy performance was good, with an average score of 70.14 (the average scores of IAA, IKS, ITB, and ISR were 21.12, 14.13, 13.36, and 21.53, respectively). However, the four dimensions of digital literacy were unbalanced. Specifically, the average scores for IAA, ITB, and ISR were more than 70%, while the average score for IKS was less than 50%. Of these, the average score for ISR was the highest (87.5%), indicating that students had commendable information ethics and exhibited a strong adherence to digital laws and regulations. The average score for IKS was the lowest (48.9%), indicating that the students had limited knowledge of information science and lacked proficiency in using digital technology to solve practical problems. The average score for ITB was 78.6%, suggesting that the students had the ability to decompose, abstract, and summarize complex problems. However, they encountered challenges in identifying the rules and characteristics of implicit information when problem-solving. The students' performance in the dimension of IAA was slightly lower than that in the ITB dimension, with an average score of 71.6%. This indicates that the students had a certain level of sensitivity and judgment to information, but they lacked the awareness of how to use digital technology to address real-world problems.

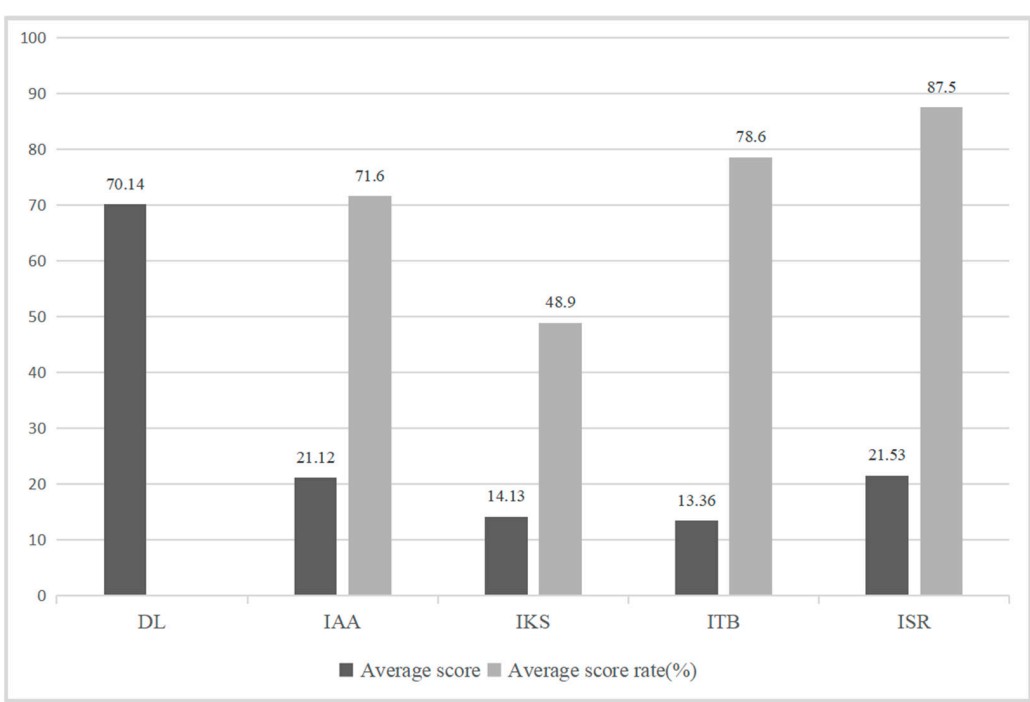

**Figure 4.** Overall results of the game-based assessment.

4.3.2. Analysis of the Feature Variables of the Game-Based Assessment

The average values of the seven feature variables of each dimension of students' digital literacy were calculated within the context of the game-based assessment tasks. It should be noted that tasks 2 and 13, relevant to ISR, were not mapped to the two feature variables of help times and answer counts. Furthermore, the tasks associated with the IAA and ISR dimensions are multiple-choice questions, which do not generate such feature variables as efficiency and similarity.

As shown in Table 4, the average values of CT and TT (from the ITB dimension) were the highest, indicating that students spent the most time (both in terms of completion time and thinking time) on the tasks relevant to the ITB dimension. In terms of the feature

variables related to game configuration, the average values of HT and AT (from the IKS dimension) were the highest, indicating that the students tended to frequently click the "Help" button and get the "Pass card", respectively. In terms of the feature variables related to behavior sequence, it is notable that only the tasks associated with the ITB and IKS dimensions encompass drag and drop questions, maze questions, connection questions, and sorting questions—all of which can collect behavior sequence data. The average values of similarity and efficiency in the ITB dimension were slightly higher than those in the IKS dimension. This implies that students exhibit a higher level of deviation between their action sequence and the reference sequence when responding to a task in terms of IKS. In terms of the feature variables related to answer results, the mean value of CO in the ISR dimension was the highest, while the CO value in the IKS dimension was the lowest.

**Table 4.** Results for the feature variables of the game-based assessment.

| Index | IAA | IKS | ITB | ISR |
|-------|-----|-----|-----|-----|
| CT | 7.468 | 18.716 | 23.758 | 6.87 |
| TT | 4.084 | 4.286 | 4.374 | 2.876 |
| CO | 71.452 | 48.168 | 77.241 | 81.422 |
| HT | 0.847 | 1.268 | 0.514 | / |
| AC | 0.769 | 1.068 | 0.375 | / |
| EF | / | 0.807 | 0.841 | / |
| SI | / | 0.587 | 0.795 | / |

*4.4. Verification of the Digital Game-Based Assessment Results*

The standardized test results were found to have a skewness of $-0.603$, a kurtosis of $-0.535$, and a Shapiro–Wilk test value of 0.948. Conversely, the game-based assessment findings had a skewness of $-0.565$, a kurtosis of $-0.576$, and a Shapiro–Wilk test value of 0.953. Given that the assessment data is non-normally distributed continuous data, this study uses Spearman's coefficient for correlation analysis. A Spearman correlation analysis was used to examine the relationship between the scores obtained from the digital literacy standardized test and the results of the digital literacy game-based assessment. The results show that the correlation coefficient was 0.918, suggesting a strong association between the results of the game-based assessment and the standardized test scores. Furthermore, this study revealed a significant association between the outcomes of the game-based assessment and the standardized test scores in connection to the four aspects of digital literacy. Specifically, the correlation coefficients for IAA, IKS, ITB, and ISR were determined to be 0.919, 0.921, 0.875, and 0.889, respectively. These results indicate that the game-based assessment tool is a highly reliable and valid instrument for assessing students' digital literacy.

**5. Discussion and Conclusions**

Working from the concept of ECGD, a digital literacy assessment for secondary school students was carried out using the digital literacy game-based assessment system developed by the research team. Specifically, this study assessed students' digital literacy through the following four steps: (1) establishing an ECGD based evaluation framework (comprising a student model, evidence model, and task model); (2) inducing students' performance related to digital literacy using the digital literacy game-based assessment system; (3) using the Delphi method to determine feature variables based on the complex and fine-grained procedural behavior data generated in the process of students' game-playing, and calculating the weights of the feature variables using the Analytic Hierarchy Process (AHP); and (4) constructing an evaluation model to measure students' digital literacy. The results show that the digital game-based assessment results were consistent with the standardized test scores, indicating that the ECGD-based digital literacy assessment approach is reliable and valid. This assessment can collect abundant evidence for digital literacy, much more so than traditional assessment methods such as standardized testing.

For example, by analyzing the similarity and efficiency of students' action sequences, one can analyze the hidden problem-solving processes behind consistent answer results, thus obtaining more objective and richer insight into students' performance. Therefore, we believe that the ECGD-based assessment method is particularly suitable for evaluating such complex and implicit competencies as digital literacy. Given the challenges of directly measuring complex and implicit competencies, it is essential to design a gamified task scenario that eliminates pressure and allows for evidence collection during gameplay to infer students' abilities.

We believe that this study makes some important theoretical and practical contributions. In terms of theory, we put forward the idea of a game-based assessment of students' digital literacy based on ECGD theory, establishing a new paradigm for the evaluation of students' digital literacy and other higher-order cognitive aspects. Specifically, the idea of a digital literacy game-based assessment based on ECGD can stimulate students' interest and induce students' digital literacy-related performance in situational tasks. Doing so effectively solves the problems affecting current research on the evaluation of students' digital literacy, such as the impact of test anxiety on the objectivity of evaluation and the difficulty of measuring students' implicit thinking ability. In practice, we have built a student digital literacy evaluation model, which provides a new way of mapping process data onto higher-order thinking ability. Specifically, this study focuses on the problem that research to date on student digital literacy evaluation focuses on result data to the detriment of process data. Thus, in this study, the game-based evaluation tool was used to collect fine-grained process data, which was then mined to determine the correlation mechanism linking the digital literacy evaluation index to the key process data. The mapping relationship and weights linking the game evaluation tasks to the characteristic variables were determined through the Delphi method and Analytic Hierarchy Process, and finally the student digital literacy evaluation model was established based on the integration of the result data with the process data, so as to reveal the underlying logic of the relationship between the two, thus accomplishing the accurate evaluation of students' digital literacy.

Based on the overall evaluation results and performance as evaluated by the feature variables of the game, this study found that the secondary school students had a moderate level of digital literacy, and the development of their four dimensions of digital literacy was not balanced. With regard to the score on each dimension, students performed better in the dimensions of IAA, ITB, and ISR, while they performed the worst in the IKS dimension. This suggests that most of the students' information consciousness and attitude are in an immature state of formation, although they have relatively keen attention and judgment to information, can use the methods in the field of computer science in problem-solving processes, and can abide by the code of ethical behavior, laws, and regulations of cyberspace. However, students' mastery of the relevant concepts, principles, and skills of information science and technology is still insufficient. Analysis of the feature variables showed that students' completion times and thinking times were longer in the ITB dimension than the other dimensions, although the score for the ITB dimension ranked second. Although the students had the ability to abstract and decompose questions, they tended to spend a longer time thinking when completing situational tasks with complex operations. Thus, it is still necessary to cultivate students' ability to discover, analyze, explore, and solve problems, so they can gradually internalize the ability to solve complex and real-world problems. In the IKS dimension, students' responses had low correctness and similarity, indicating that their mastery of scientific knowledge and application skills was not strong. Furthermore, the students were more inclined to click the "Help" button while completing the tasks associated with the IKS dimension. These findings address the necessity and significance of teaching the basic knowledge and skills of information science and technology, which should be the foundation of students' digital literacy education. In the ISR dimension, students not only achieved their highest scores, but also had the shortest completion times and thinking times. This suggests that students possess the ability to swiftly make appropriate decisions regarding digital ethics and morality. Consequently, it can be inferred that students possess

a strong awareness in areas such as information security, risk management, and control, as well as intellectual property protection. Overall, the aforementioned research findings align with the outcomes of the large-scale standardized assessment results conducted by the research team in the early stage [65].

This study has some limitations that should be noted. First, its case size was relatively small, and only one school was selected for assessment. We plan a follow-up study that will expand the case size to collect cases of primary and secondary school students from all regions of China, so as to verify the effectiveness of the ECGD-based digital game assessment approach with more and richer data, and improve the reliability and representativeness of the research results. Second, this study only collected the clickstream data of students in the game assessment tasks, but did not collect eye movement, video, or other multimodal data. Future studies may make use of multimodal data, such as facial expression, facial posture, and eye movement, all of which could reflect students' performance in terms of digital literacy. Finally, although this study compared the game-based assessment results with standardized test scores via the Spearman correlation analysis, the ECGD-based assessment approach should be further verified in future research with a psychological measurement model, such as the cognitive diagnosis model.

**Author Contributions:** Writing, methodology, software, J.L.; assessment design, writing, J.B.; conceptualization, writing review and editing, S.Z.; supervision, H.H.Y. All authors have read and agreed to the published version of the manuscript.

**Funding:** The work was funded by a grant from the National Natural Science Foundation of China (No. 62107019), the Key project of the special funding for educational science planning in Hubei Province in 2023 (2023ZA032), and the Key Subjects of Philosophy and Social Science Research in Hubei Province of 2022 (22D043).

**Institutional Review Board Statement:** Ethics Review Committee, Faculty of Artificial Intelligence in Education, Central China Normal University stated the procedures for human participants involved in this study were consistent with the ethical standards of Central China Normal University and the 1975 Helsinki Declaration. The approval number is CCNU-IRB-202204002b. The approved date is 28 March 2022.

**Informed Consent Statement:** Prior to participation, all participants were provided with information regarding the study's purpose and were required to sign a informed consent form.

**Data Availability Statement:** The datasets presented in this article are not readily available given the confidential nature of the data. Requests to access the datasets should be directed to zhusha@mail.ccnu.edu.cn.

**Acknowledgments:** This article is an extended version of "Assessing secondary students' digital literature using an Evidence-Centered Game Design approach", which was published in the 16th International Conference on Blended Learning (pp. 214–223) by Springer Nature Switzerland. We appreciate the recommendation of the conference and the permission of the authors to publish the extended version.

**Conflicts of Interest:** The authors declare that there are no potential conflicts of interest.

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
