# Peer review of "Game-Based Assessment of Students’ Digital Literacy Using Evidence-Centered Game Design†"

_electronics, doi:10.3390/electronics13020385_

Round 1

Reviewer 1 Report

Comments and Suggestions for Authors

The paper reports on an interesting game-based method for the assessment of digital literacy. The paper is well written, and the presented results are methodologically sound. Some points that the authors should address in the paper are the following:

It is mentioned on p. 4, line 174 that game-based assessment can reduce anxiety in the assessment process. However, the proposed method uses thinking and answering times as measurements of performance. Thus, it is likely that anxiety may be increased instead of reduced during assessment. Thus, how is the proposed method related to anxiety reduction, compared to traditional forms of assessment?

It should be discussed whether  the proposed assessment method can be applied in domains other than digital literacy. 

The authors should explicitly define the acronym ECD on p. 4, line 157 (possibly, Evident-Centered Design).

Author Response

Reviewer #1:

The paper reports on an interesting game-based method for the assessment of digital literacy. The paper is well written, and the presented results are methodologically sound. Some points that the authors should address in the paper are the following:

  1. It is mentioned on p. 4, line 174 that game-based assessment can reduce anxiety in the assessment process. However, the proposed method uses thinking and answering times as measurements of performance. Thus, it is likely that anxiety may be increased instead of reduced during assessment. Thus, how is the proposed method related to anxiety reduction, compared to traditional forms of assessment?

Response: Thank you for your insightful comment. During game-based assessment tasks, students are unaware that their response time and other data will be used to assess digital literacy. To them, it is merely an engaging activity, potentially alleviating test anxiety compared to traditional question-answer formats.

  1. It should be discussed whether the proposed assessment method can be applied in domains other than digital literacy. 

Response: Thank you for your valuable input. We mentioned in the manuscript’s, discussion, “we believe that the ECGD-based assessment method is particularly suitable for evaluating such complex and implicit competencies as digital literacy.” To elaborate on this, we expanded our discussion in this section. The revised content is as follows:

Therefore, we believe that the ECGD-based assessment method is particularly suitable for evaluating such complex and implicit competencies as digital literacy. Given the challenges of directly measuring complex and implicit competencies, it is essential to design a gamified task scenario that eliminates pressure and allows for evidence collection during gameplay to infer students’ abilities.

  1. The authors should explicitly define the acronym ECD on p. 4, line 157 (possibly, Evident-Centered Design).

Response: Thank you for pointing this out. We defined ECD for the first time in part 2.4 as follows:

Mislevy proposed the ECGD approach [38]. Based on Evidence-Centered Design (ECD), this approach incorporates game-based tasks into assessment design.

Reviewer 2 Report

Comments and Suggestions for Authors

Dear Authors,

I find your research on game-based assessment of students’ digital literacy interesting and insightful especially for the educational practices. However, for a more comprehensive and publishable paper, I suggest several minor adjustments:

  • Literature review – Subsection 2.1. Digital literacy: after presenting the three stages, it seems that the following paragraph (lines 122-128) does not sound like a continuation of the previous ones. Please try to develop the ideas and make a good transition to it.
  • Line 157: the acronym ECD was not previously introduced. Please use the full concept when referring to it for the first time in your text, as you did with the other acronyms.
  • Subsection 3.5.: please consider replacing ”samples” with “cases” for a better understanding.
  • Line 370: “As shown in Fig. 3” – should be replaced with “As shown in figure 2” – the complete word, “figure”, and number 2 instead of 3.
  • Line 382: “(see Figure 2)” – it should be replaced with “(see figure 3)”.
  • Figure 3 – for clarity, you should keep only the average score rate (%) – the ones that you also interpreted in the text. The others are not clear to the reader (of course, I can check and see that 70.14 is the sum of the other four, but this does not help me better understand the results).
  • Results – Subsection 4.4. (lines 423-433): please expand a bit this part. First, show the descriptive statistics for the variables used (in your case scores obtained from the digital literacy standardized test and the results of the digital literacy game-based assessment, IAA, IKS, ITB, and ISR), by providing also data about skewness, kurtosis and p-value Shapiro-Wilk, to test the normality of the distributions. Choose Pearson if the data is normal, or else opt for a nonparametric equivalent like Spearman or Kendall’s tau-b.
  • Language: I am not a native speaker of English, so you should discuss this with somebody with higher expertise from this point of view. However, for more clarity, I recommend changing During gameplay, students different gain gold coins according to their task performance.” (lines 223-224) to something like: During gameplay, students gain gold coins based on their task performance.” Then, in line 349 you should remove „s” at the end of the word „represents” => „represent”. Also, at first view for me it was not clear the answer times “(whether students completed the task successfully on the first try)” (lines 232-233) – in the sense that the word “whether” shows something like yes or no, while, for me answer time could have a lot more options, not only two. I understood only when reading lines 364-366 that it is still binary. That variable is more like a First_Answer_Correct (but maybe you could find an even better name).

I hope these suggestions will help you refine your paper, making it more cohesive and accessible to a wider audience.

Good luck with your work!

Author Response

Reviewer #2:

I find your research on game-based assessment of students’ digital literacy interesting and insightful especially for the educational practices. However, for a more comprehensive and publishable paper, I suggest several minor adjustments:

  1. Literature review – Subsection 2.1. Digital literacy: after presenting the three stages, it seems that the following paragraph (lines 122-128) does not sound like a continuation of the previous ones. Please try to develop the ideas and make a good transition to it.

Response: Thank you. In response to your comment, we have summarized and concluded the conceptual connotation of the three stages, and put forward our definition of digital literacy on this basis. To maintain clarity, we have relocated the content regarding the digital literacy framework into Section 2.2.

The revised content (please see the red text below) in Section 2.1 is as follows:

In summary, from the conceptual development of digital literacy, the connotation of digital literacy has expanded from only including understanding, using, assessing, and integrating digital resources to innovation and creativity, ethics and morality, etc., and from only proposing cognitive level requirements such as awareness and knowledge to behavioral and value level requirements [21]. In addition, with the continuous develop-ment of artificial intelligence, digital literacy ultimately includes a wider range of high-er-order content, such as information thinking. The concept of digital literacy has become more comprehensive and essential for individuals to thrive in the digital age. Specifically, digital literacy includes low-level cognitive aspects such as basic information knowledge, as well as high-level cognitive aspects such as using information technology to solve problems. Based on the above points of view, our research team posits that digital literacy is a multifaceted construct comprising an individual’s awareness, ability, thinking and cultivation to properly use information technology to acquire, integrate, manage, and evaluate information; understand, construct, and create new knowledge; and to discover, analyze, and solve problems [22].

  1. Line 157: the acronym ECD was not previously introduced. Please use the full concept when referring to it for the first time in your text, as you did with the other acronyms.

Response: Thank you. Your comment is very helpful. We defined ECD in section 2.4 for the first time:

Mislevy proposed the ECGD approach [38]. Based on Evidence-Centered Design (ECD), this approach incorporates game-based tasks into assessment design.

  1. Subsection 3.5.: please consider replacing ”samples” with “cases” for a better understanding.

Response: Thank you. To improve clarity, we have replaced the term “samples” with “cases”.

  1. Line 382: “(see Figure 2)” – it should be replaced with “(see figure 3)”.

Response: Thank you very much. We apologize for this mistake, which has now been corrected.

  1. Figure 3 – for clarity, you should keep only the average score rate (%) – the ones that you also interpreted in the text. The others are not clear to the reader (of course, I can check and see that 70.14 is the sum of the other four, but this does not help me better understand the results).

Response: Thank you. In response to your comment, we have removed the average score rate of DL but have retained the average scores of the four dimensions as they serve as the basis for calculating DL. Additionally, we have included text to explain the origin of DL average score.

The revised content (please see the red text below) in Section 4.3.1 is as follows:

The results (see Figure 4) show that the students’ overall digital literacy performance was good, with an average score of 70.14 (the average scores of IAA, IKS, ITB and ISR were 21.12, 14.13, 13.36 and 21.53, respectively).

  1. Results – Subsection 4.4. (lines 423-433): please expand a bit this part. First, show the descriptive statistics for the variables used (in your case scores obtained from the digital literacy standardized test and the results of the digital literacy game-based assessment, IAA, IKS, ITB, and ISR), by providing also data about skewness, kurtosis and p-value Shapiro-Wilk, to test the normality of the distributions. Choose Pearson if the data is normal, or else opt for a nonparametric equivalent like Spearman or Kendall’s tau-b.

Response: Thank you for your valuable comment. We appreciate your reminder. After conducting a normal distribution test on both the standardized test and game-based assessment test data, we found that they did not conform to a normal distribution. As a result, we opted to use Spearman correlation analysis instead.

The revise content (please see the red text below) in section 4.4 is as follows:

The standardized test results were found to have a skewness of -0.603, a kurtosis of -0.535, and a Shapiro-Wilk test value of 0.948. On the other hand, the game-based as-sessment findings had a skewness of -0.565, a kurtosis of -0.576, and a Shapiro-Wilk test value of 0.953. Given that the assessment data is non-normally distributed continuous data, this study uses Spearman's coefficient for correlation analysis. Spearman correlation analysis was used to examine the relationship between the scores obtained from the digital literacy standardized test and the results of the digital literacy game-based assessment.

  1. Language: I am not a native speaker of English, so you should discuss this with somebody with higher expertise from this point of view. However, for more clarity, I recommend changing “During gameplay, students different gain gold coins according to their task performance.” (lines 223-224) to something like: “During gameplay, students gain gold coins based on their task performance.” Then, in line 349 you should remove „s” at the end of the word „represents” => „represent”. Also, at first view for me it was not clear the answer times “(whether students completed the task successfully on the first try)” (lines 232-233) – in the sense that the word “whether” shows something like yes or no, while, for me answer time could have a lot more options, not only two. I understood only when reading lines 364-366 that it is still binary. That variable is more like a First_Answer_Correct (but maybe you could find an even better name).

Response: Thank you very much for your thorough and helpful comments. We have made revisions based on your suggestions. Firstly, we have changed “during game play, students different gain gold coins according to their task performance” to “during game play, students gain gold coins based on their task performance”. Second, we have changed “represents” to “represent”. Third, we have changed “answer times” to “answer counts”, and its expression to “number of times of a task completion”.

Reviewer 3 Report

Comments and Suggestions for Authors

This study focuses on digital game-based digital literacy assessment method, a timely and important topic. This study raises some fundamental questions that need to be answered prior to its consideration for publication:

First, since this study is about game-based assessment, why did you organize another validated test? Isn’t the game sufficient for this purpose?

Second, the sequence of data collection phases should be justified (see also relevant point below).

Additional issues:

L33-40: These are strong statements on current digital literacy assessment methods. Is there literature evidence that supports and confirms them?

L43-47: Interesting observation but not straightforward to the reader. What are the reasons behind this failure of situational task assessment and portfolio assessment methods?

L50: Please explain each acronym (ECGD) on its first appearance in the main text as well.

L61-66: Please examine if this information is necessary here, it could be moved to the Method section and explained in more depth.

L129: Section 2.1 review is comprehensive. However, in section 2.1 or 2.2 it could be useful to add a section on digital literacy frameworks e.g. [1].

The analysis of what digital literacy entails should be linked with the assessment of different digital literacy competence skills and levels, otherwise section 2.2 would remain in limbo: what knowledge or skills or domain is assessed exactly and of which audience? Digital literacy of citizens could differ in scope from higher education students or professionals.

L161: What is the difference between ECD and other methods such as CTT?

L176: Authors mention several times the concept “higher-order thinking abilities”. First, it would be useful to explain what this entails exactly in this context.

Second, while higher-order deep thinking abilities are essential in education [2,3] please explain why this concept is relevant for digital literacy.

L221: What are these four dimensions of digital literacy?

L222: Please provide examples of these tasks.

Table 1: Please explain each dimension of digital literacy beforehand. These acronyms are incomprehensible to the reader.

Are 12 tasks sufficient for such a comprehensive assessment or provide just a glimpse of a person’s abilities? Please provide evidence that validates task design choices.

L229: Why is completion time relevant? If a task is completed correctly, why does speed matter for his/her assessment?

L233: Since the purpose of the game is assessment, what is the purpose of a help button? If a student is unable to complete a task, should s/he just move on and record his current competence level?

L245: Please provide more information about the test and some examples, readers should not be forced to read another article to understand how this process was implemented.

L275: Why was “data collected in this study redundant, inconsistent, and even noisy”? Please explain what caused the phenomenon of missing data.

L288: Why was the Delphi method organized at that order? One would expect this to take place during the design and pilot testing phase of the game to ensure task and assessment validity.

Please create a flow diagram with all phases of the research study.

1.            ISBN 978-92-79-73494-6

2.            doi:10.3390/encyclopedia1030075

3.            ISBN 978-1-61735-505-9

Author Response

Reviewer #3:

This study focuses on digital game-based digital literacy assessment method, a timely and important topic. This study raises some fundamental questions that need to be answered prior to its consideration for publication:

  1. First, since this study is about game-based assessment, why did you organize another validated test? Isn’t the game sufficient for this purpose?

Response: Thank you. This study introduces a game-based assessment method, but its reliability and validity cannot be confirmed without further verification. To evaluate the effectiveness of the proposed method, correlation analysis was employed to examine the relationship between the results of the game-based assessment and the standardized test. The rationale for conducting both tests has been added in Section 3.5.

The added content (please see the red text below) in Section 3.5 is as follows:

Finally, in order to verify the effectiveness of the game-based digital literacy assessment method proposed in this study, Spearman correlation analysis was conducted to analyze the correlation between the game-based assessment results and the standardized test results. The standardized test utilized by this research team has been applied in multiple large-scale assessments, confirming its reliability and validity as a tool for evaluating students’ digital literacy [48]. Therefore, the correlation between the game-based assessment results and the standardized test results could validate the effectiveness of the assessment method proposed in this study.

  1. Second, the sequence of data collection phases should be justified (see also relevant point below).

L288: Why was the Delphi method organized at that order? One would expect this to take place during the design and pilot testing phase of the game to ensure task and assessment validity.

Please create a flow diagram with all phases of the research study.

Response: Thank you for your comment. Actually, our research is currently in the design and pilot testing phase. The aim of this study is to employ the Delphi method and AHP to identify characteristic variables based on test data, and to validate this approach. In the future, we plan to conduct large-scale assessments using the approach proposed in this study.

According to your suggestion, we have streamlined the data analysis procedure of this study by adding and revising related content in Section 3.5. It should be noted that, we choose to described the five steps of data analysis in sequential order, which might be more appropriate, instead of adding a flow diagram.

The added and revised content (please see the red text below) in Section 3.5 is as follows:

The data analysis phase includes the following five steps: 1) determined the feature variables of students’ digital literacy based on Delphi method; 2) calculated the weights of the feature variables using the AHP; 3) analyzed the game-based assessment data using a Rasch model to validate the task design; 4) analyzed the results of the game-based assessment for students’ digital literacy; 5) verify the effectiveness of the game-based digital literacy assessment method using Spearman correlation analysis.

Firstly, the Delphi method was used to determine the feature variables and construct the evaluation model of students’ digital literacy. An “expert advisory group” was established, composed of 14 scholars, teachers, and researchers in the field of digital literacy assessment. They were invited to complete a survey, determine the feature variables of each task in relation to students’ digital literacy, and rank these feature variables according to their importance. Specifically, this study conducted three rounds of expert consultations. In each round, a questionnaire was distributed to the experts and their feedback was collected, analyzed, and summarized. The feature variables were revised and iterated based on the experts’ opinions. After the third round of consultation, the experts’ opinions were generally consistent and met the expected requirements.

Secondly, the weights of the characteristic variables were determined using the AHP. The analysis steps of AHP were as follows: 1) a judgment matrix was built by comparing the importance of the characteristic variables; 2) the weight of the characteristic variable of each task was calculated; and 3) the consistency of the judgment matrix was verified [49]. If the consistency of the judgment matrix was verified, then, these weights were served as the scoring criteria for measuring students’ digital literacy. Otherwise, the researcher would delete the questionnaire data that are inconsistent with the judgment matrix and recalculate the weights.

Thirdly, in order to explore the relationship between item difficulty and students’ ability, this study generated an item-person map using a Rasch model. This is a graphical representation of person-abilities and item-difficulties, drawn based on the equal measures (logits) of the raw item difficulties and raw person scores [50]. The item-person map is divided into four quadrants, in which person estimates and item estimates are distributed on the left and right sides, respectively, based on person-ability and item-difficulty estimates [51]. Generally, the persons in the upper left quadrant show better abilities, implying that the easier items were not difficult enough for them. Mean-while, the items on the upper right show higher difficulty, suggesting that they are be-yond the students’ ability level [11]. Rasch model analysis includes the following three steps: 1) computing the unidimensionality; 2) calculating person and item reliability coefficients; and 3) generating an item-person map [52].

Fourthly, the average scores for students’ overall digital literacy and its four individual dimensions, as well as the average values of the seven feature variables of each dimension of students’ digital literacy were calculated within the context of the game-based assessment tasks.

Finally, in order to verify the effectiveness of the game-based digital literacy assessment method proposed in this study, Spearman correlation analysis was conducted to analyze the correlation between the game-based assessment results and the standardized test results. The standardized test utilized by this research team has been applied in multiple large-scale assessments, confirming its reliability and validity as a tool for evaluating students' digital literacy [48]. Therefore, the correlation between the game-based assessment results and the standardized test results could validate the effectiveness of the assessment method proposed in this study.

  1. L33-40: These are strong statements on current digital literacy assessment methods. Is there literature evidence that supports and confirms them?

Response: Thank you. In response to your comments, we have added references to support the statements in the section of Introduction.

The added references in the section of Introduction are as follows:

First, evaluation content to date has primarily focused on traditional “test question-answer” paradigms [5], emphasizing students’ basic cognitive abilities such as digital knowledge and application skills. Consequently, such an approach fails to evaluate higher-order thinking skills, such as the ability to use digital technology to analyze and solve problems, engage in innovation and creativity, and address ethical and moral issues in digital society [6].

This revision included the addition of the following reference:

[5] Wu, D., Zhou, C., Li, Y., & Chen, M. (2022). Factors associated with teachers’ competence to develop students’ information literacy: A multilevel approach. Computers & Education, 176, 104360.

[6] Chen, M., Zhou, C., Meng, C., & Wu, D. (2019). How to promote Chinese primary and secondary school teachers to use ICT to develop high-quality teaching activities. Educational Technology Research and Development, 67, 1593-1611.

  1. L43-47: Interesting observation but not straightforward to the reader. What are the reasons behind this failure of situational task assessment and portfolio assessment methods?

Response: Thank you. To enhance readers’ understanding, we have added content to address the problems associated with situational task assessment and portfolio assessment, along with supporting references.

The added content (please see the red text below) in the section of introduction is as follows:

Although several studies have introduced situational task assessment and portfolio assessment as alternative tools for process evaluation in recent years, the process data collected by the situational task assessment is of single type and coarse granularity, which is relatively inadequate in providing comprehensive evidences reflecting students’ digital literacy [9]. Establishing scientific and objective evaluation standards for portfolio assessment is challenging, and it requires teachers to invest a great deal of time and effort, which limits the application of portfolio assessment in evaluating students’ digital literacy [10]. Therefore, these two assessment methods could not accurately evaluate students' digital literacy [11].

  1. L50: Please explain each acronym (ECGD) on its first appearance in the main text as well.

Response: Thank you. We have clarified the term ECGD when it first appears in the text.

The revised content (please see the red text below) is as follows:

The application of the Evidence-Centered Game Design (ECGD) approach in assessment design has the potential to overcome the limitations of the current methods for the evaluation of students’ digital literacy.

  1. L61-66: Please examine if this information is necessary here, it could be moved to the Method section and explained in more depth.

Response: Thank you. Based upon your comments, we have simplified the description of methods in the introduction, and made an in-depth exploration in Section 3.5.

The revised content (please see the red text below) in the section of Introduction is as follows:

The Delphi method and the Analytic Hierarchy Process (AHP) are employed to determine the extracted characteristic variables and their weights. Furthermore, the research team has conducted a practical analysis to validate the proposed method.

The revised content (please see the red text below) in Section 3.5 is as follows:

The analysis steps of AHP were as follows: 1) a judgment matrix was built by comparing the importance of the characteristic variables; 2) the weight of the characteristic variable of each task was calculated; and 3) the consistency of the judgment matrix was verified [49]. If the consistency of the judgment matrix was verified, then, these weights were served as the scoring criteria for measuring students’ digital literacy. Otherwise, the researcher would delete the questionnaire data that are inconsistent with the judgment matrix and recalculate the weights.

  1. L129: Section 2.1 review is comprehensive. However, in section 2.1 or 2.2 it could be useful to add a section on digital literacy frameworks e.g. [1].

The analysis of what digital literacy entails should be linked with the assessment of different digital literacy competence skills and levels, otherwise section 2.2 would remain in limbo: what knowledge or skills or domain is assessed exactly and of which audience? Digital literacy of citizens could differ in scope from higher education students or professionals.

Response: Thank you. According to your suggestion, we have added a new section (section 2.2) to present digital literacy framework.

The added Section 2.2 is as follows:

At present, international organizations and researchers pay great attention to the assessment of students’ digital literacy, and has conducted extensive and in-depth research on the establishment of the assessment framework of students’ digital literacy. For example, European Commission (EC) proposed DigComp1.0 [23] in 2013, including five areas of digital competence: information, communication, content-creation, safety, and problem-solving. In 2016, EC proposed DigComp2.0 on the basis of DigComp1.0 [24], including five competence areas: information and data literacy, communication and collaboration, digital content creation, safety, and problem solving. UNESCO released a digital literacy global framework [25] in 2018, adding two new dimensions “hardware and software operations” and “career related competencies” on the basis of DigComp2.0. Additionally, in terms of researchers, Eshet-Alkalai proposed a digital literacy framework based on years of research and practical experience, including: photo-visual literacy, re-production literacy, branching literacy, information literacy and socio-emotional literacy [16].

Based upon the review of the digital literacy framework proposed by international organizations and researchers, the research team extracted, sorted and merged the key words of the assessment indicators of students’ digital literacy. Thus, the assessment framework of students’ digital literacy is proposed, which includes four dimensions: in-formation awareness and attitude (IAA), information knowledge and skills (IKS), information thinking and behavior (ITB), and information social responsibility (ISR) [26]. IAA refers to the sensitivity, judgment and protection awareness of information privacy and security, including information perception awareness, information application awareness and information security awareness; IKS mainly investigates students’ understanding of information science knowledge, mastering and using common software or tools to complete the creation of digital works, mainly involving information science knowledge and information application skills; ITB mainly refers to the thinking ability to abstract, decompose and design problems in daily life and learning situations, as well as the comprehensive consciousness and ability tendency to carry out digital communication and learning, mainly including two levels of information thinking and information behavior; ISR refers to the responsibility of students’ activities in the information society in terms of laws, regulations and ethics, including information ethics and laws and regulations [22].

  1. L161: What is the difference between ECD and other methods such as CTT?

Response: Thank you. According to your suggestion, we have added content explaining the difference between ECD and other methods such as CTT.

The added content (please see the red text below) in Section 2.4 is as follows:

Mislevy proposed the ECGD approach [38]. Based on Evidence-Centered Design (ECD), this approach incorporates game-based tasks into assessment design. ECD is a systematic method that guides the design of educational evaluations [39]. Compared to traditional assessment methods such as CTT that assign values to potential traits based on features, ECD may collect more extensive and fine-grained process data from students, which allows for comprehensive and accurate evaluation of their complex implicit abilities. Specifically, ECD emphasizes the construction of complex task situations, obtaining multiple types of procedural data, and achieving evidence-based reasoning [40].

  1. L176: Authors mention several times the concept “higher-order thinking abilities”. First, it would be useful to explain what this entails exactly in this context.

Second, while higher-order deep thinking abilities are essential in education [2,3] please explain why this concept is relevant for digital literacy.

Response: Thank you. In order to avoid confusion, we have changed “higher-order thinking abilities” to “higher-order cognitive aspects”, and added a related elaboration in Section 2.1 to explain to the readers the specific content of the lower-order cognitive aspects and higher-order cognitive aspects of digital literacy.

The added content (please see the red text below) in Section 2.1 is as follows:

In addition, with the continuous development of artificial intelligence, digital literacy ultimately includes a wider range of higher-order content, such as information thinking. The concept of digital literacy has become more comprehensive and essential for individuals to thrive in the digital age. Specifically, digital literacy includes lower-order cognitive aspects such as basic information knowledge, as well as higher-order cognitive aspects such as using information technology to solve problems. Based on the above points of view, our research team posits that digital literacy is a multifaceted construct comprising an individual’s awareness, ability, thinking and cultivation to properly use information technology to acquire, integrate, manage, and evaluate information; understand, construct, and create new knowledge; and to discover, analyze, and solve problems [22].

  1. What are these four dimensions of digital literacy?

Table 1: Please explain each dimension of digital literacy beforehand. These acronyms are incomprehensible to the reader.

Response: Thank you. According to your suggestion, we have explained each dimension of digital literacy in Section 2.2.

The added content (please see the red text below) regarding with the four dimensions is as follows:

Based upon the review of the digital literacy framework proposed by international organizations and researchers, the research team extracted, sorted and merged the key words of the assessment indicators of students’ digital literacy. Thus, the assessment framework of students’ digital literacy is proposed, which includes four dimensions: information awareness and attitude (IAA), information knowledge and skills (IKS), information thinking and behavior (ITB), and information social responsibility (ISR) [26]. IAA refers to the sensitivity, judgment and protection awareness of information privacy and security, including information perception awareness, information application awareness and information security awareness; IKS mainly investigates students’ understanding of information science knowledge, mastering and using common software or tools to complete the creation of digital works, mainly involving information science knowledge and information application skills; ITB mainly refers to the thinking ability to abstract, decompose and design problems in daily life and learning situations, as well as the comprehensive consciousness and ability tendency to carry out digital communication and learning, mainly including two levels of information thinking and in-formation behavior; ISR refers to the responsibility of students' activities in the information society in terms of laws, regulations and ethics, including information ethics and laws and regulations [22].

  1. L222: Please provide examples of these tasks.

Response: Thank you for your suggestion. We have expanded the content to include more information about the 13 tasks and provided a sample task for reference.

The revised content (please see the red text below) regarding with the tasks is as follows:

In order to induce performance relative to the students’ digital literacy ability model, this study uses the narrative game “Guogan’s Journey to A Mysterious Planet”, developed by our research team to evaluate students’ digital literacy [47]. The game consists of a total of 13 tasks measuring the four dimensions of students’ digital literacy, which can be categorized into five different types: multiple-choice question, maze, dragging question, matching question, and sorting question. Each task was designed to measure one dimension of students’ digital literacy. For instance, Figure 1 shows a dragging question, which requires the students to discover patterns or rules by analyzing the existing information, and drag the correct color block to the right position. This task examines the ITB dimension of students’ digital literacy. Students are given two chances to complete each task. If a student finishes a task incorrectly twice, the system presents a “Pass card” and begins the next task. During game-play, students gain gold coins based on their task performance. Students can choose whether to click a “Help” button during the gameplay process; doing so costs them coins, but it will provide students with some hints to complete the tasks successfully. They can also click a “Return” button to return to the previous page to confirm information.

  1. Are 12 tasks sufficient for such a comprehensive assessment or provide just a glimpse of a person’s abilities? Please provide evidence that validates task design choices.

Response: Thank you. The 13 tasks utilized in this study span across four dimensions of students' digital literacy, with each task assessing a distinct dimension, which enables a comprehensive evaluation of students’ digital literacy. To clarify this point, we have added related content in Section 3.2.1. Furthermore, the findings from the Rasch model analysis suggest that the task design is effective and appropriate for measuring students’ digital literacy.

Section 4.2 presented specific modifications:

In order to induce performance relative to the students’ digital literacy ability model, this study uses the narrative game “Guogan’s Journey to A Mysterious Planet”, developed by our research team to evaluate students’ digital literacy [47]. The game consists of a total of 13 tasks measuring the four dimensions of students’ digital literacy, which can be categorized into five different types: multiple-choice question, maze, dragging question, matching question, and sorting question. Each task was designed to measure one dimension of students’ digital literacy. 

  1. L229: Why is completion time relevant? If a task is completed correctly, why does speed matter for his/her assessment?

Response: Thank you. Through extensive literature review, we found that the time students spend on completing tasks reflects their proficiency in completing tasks and is used as a key indicator of students’ ability. Furthermore, after consulting with experts, we discovered that the majority of them believe that completion time is a very important feature variable in evaluating students’ digital literacy within the context of the game-based assessment system. Thus, we have added content in Section 3.2.1 to explain the rationale for selecting feature variables, including completion time.

The added content (please see the red text below) in Section 3.2.1 is as follows:

Table 1 shows the details of the game-based assessment tasks, including the task type, the corresponding dimension of digital literacy, and the observed variables. These observable variables have been considered as evidences that can reflect the performance of students' digital literacy through extensive literature reviews and expert consultation [47].

  1. L233: Since the purpose of the game is assessment, what is the purpose of a help button? If a student is unable to complete a task, should s/he just move on and record his current competence level?

Response: Thank you. The “Help” button can provide students with task-related hints during the task completion process to help them successfully complete the task and reduce anxiety. To further clarify this point, we have included additional explanation in Section 3.2.1.

The added content (please see the red text below) in Section 3.2.1 is as follows:

Students can choose whether to click a “Help” button during the gameplay process; doing so costs them coins, but it will provide students with some hints to complete the tasks successfully.

  1. L245: Please provide more information about the test and some examples, readers should not be forced to read another article to understand how this process was implemented.

Response: Thank you. Based upon your suggestion, we have provided a sample question of the standardized test in Section 3.2.2.

The added content (please see the red text below) in Section 3.2.2 is as follows:

To verify the results of the game-based assessment test, this study used the digital literacy standardized test designed in our previous study, comprising 24 multiple-choice questions and multiple-answer questions (i.e., multiple choice questions with more than one correct answer). For example, the following is a sample question measuring students’ IKS: The vehicle position tracker can determine the location of the vehicle in real time. What technology does it use? A. IoT technology; B. big data technology; C. cloud computing technology; D. multimedia technology. This test has been validated many times in China’s large-scale student digital literacy assessment project, and has been demonstrated to have good reliability and validity, difficulty, discrimination, and other indicators, with high standard values [48].

  1. L275: Why was “data collected in this study redundant, inconsistent, and even noisy”? Please explain what caused the phenomenon of missing data.

Response: Thank you. We have added content to explain what caused the phenomenon of missing data in Section 3.5.

The added content (please see the red text below) in Section 3.5 is as follows:

The game-based assessment data of 210 students were collected and matched with their standardized test scores based on the xAPI-based process data. During the assessment, a few students made mistakes or did not follow the operation rules of the game (For example, students accidentally exited the assessment system interface and re-logged in to continue, resulting in abnormal data), which led to the data collected in this study were redundant, inconsistent, and even noisy, it was difficult to meet the experimental requirements. Therefore, the data were cleaned and preprocessed as follows. 1) Missing data processing. When the proportion of missing student data cases was relatively high, it was necessary for the data to be eliminated. In this study, the cases with a high rate of missing fields were eliminated, and the cases with low missing rates were filled by the mean method. A total of 10 cases were eliminated for this reason. 2) Abnormal data processing. Abnormal data were those generated by students who failed to adhere to the assessment rules for their operations, as this has an impact on the accuracy of the model prediction. A total of 12 cases were eliminated due to human errors, repeated answers by the same student, or too short of a response time. Ultimately, a total of 22 cases were eliminated, and the remaining 188 cases were used as experimental data.

Round 2

Reviewer 3 Report

Comments and Suggestions for Authors

Authors addressed all identified issues in a satisfactory manner and improved the quality of their submission rendering the manuscript fit for publication. Congratulations.

Author Response

Thank you very much for your constructive suggestions.